# OpenReview forum: "On Fairness of Low-Rank Adaptation of Large Models"
_colmweb.org/COLM/2024/Conference — COLM_

### Official Review · Reviewer_i2tt · 2024-05-09

**Rating:** 7
**Confidence:** 5
**Ethics Flag:** 1

**Summary:**

This work did a comprehensive benchmark of fairness for models fine-tuned by low-rank adaptation (LoRA). The models are evaluated on both classification and natural language tasks.

---
After reading the authors response, I will retain my rating. For improving the current work, I would recommend the authors to evaluate the LoRA models enforced with fairness constraints.

**Questions To Authors:**

1. Is the model fine-tuned subject to fairness constraints? If not, I would like to see the comparison to the LoRA models that are explicitly enforced with fairness constraints.

**Reasons To Accept:**

The evaluation results are technically solid. The paper is well written. The paper could attract a large number of audience interested in the fairness of large models.

**Reasons To Reject:**

My own observation is that there is no significant difference of fairness metrics between LoRA fine-tuned and fully fine-tuned models. I don't think such observation implies anything on whether LoRA may exacerbate the biases or not, given the fact that the learning objectives are still identical.

---

> ### Author Rebuttal · Authors · 2024-05-31
>
> We thank the reviewer for their time and are glad that the reviewer finds our work valuable! We respond to the main comments below:
>
> > My own observation is that there is no significant difference of fairness metrics between LoRA fine-tuned and fully fine-tuned models. I don't think such observation implies anything on whether LoRA may exacerbate the biases or not, given the fact that the learning objectives are still identical.
>
> We agree with the reviewer that our study provides *no evidence* for LoRA exacerbating unfairness (i.e. our study, albeit comprehensive, can possibly, but unlikely, be a false negative) and this is indeed distinct from *having evidence* that LoRA does *not* exacerbate fairness. We have acknowledged this limitation in Sec 4 and will ensure this is clear in the updated version.
>
> > Is the model fine-tuned subject to fairness constraints?
>
> No, both LoRA and full fine-tuning are performed with standard Adam. We agree that it would be interesting to see how fair training (e.g. fairness constraints, or fair learning algorithms) will influence the fairness results. However, as we discussed at the end of section 2, since our end goal is to compare LoRA with full fine-tuning, we believe that the fine-tuning task *should not teach* the model to be fair, since it would otherwise introduce confounding factors of whether LoRA or full FT *responds* to a particular fairness constraint or fair learning algorithm better. Teaching the model to be fair would make whatever fairness conclusions we draw less generalizable to unseen tasks and models.
>
> Nevertheless, the reviewer has raised a good point and we hope to add relevant discussions in the updated version.

---

### Official Review · Reviewer_tTvn · 2024-05-10

**Rating:** 3
**Confidence:** 3
**Ethics Flag:** 1

**Summary:**

The paper aims to investigate the relationship between Low-Rank adaptation and the fairness of language models. Fairness is an extremely important factor in a language model, especially if these language models are used for important decisions.

**Questions To Authors:**

Provide a good reason why Low-Rank adaptation should have any influence on fairness of language models. See Reasons to Reject.

**Reasons To Accept:**

The paper addresses fairness of Language Models.

**Reasons To Reject:**

The main research question is really unclear: the relation between Low-Rank adaptation and fairness is really questionable. Let's use a metaphor. Suppose the goal is to prepare a system that is fair in showing traffic jams by hiding plates. Clearly, this kind of system is expected to blur only plates and keep everything in a clear shape. Now, it is clear that if the system blurs the whole picture, it blurs the plates too. The goal of hiding the plates is accomplished but final images are useless. Now, out of the metaphor, low-ranking is acting indiscriminately over all the parameters. So, it is clear that it has effects on every aspect of LMs. Then, why focusing on fairness? Is there any reason why this low-ranking should act only on fairness?

---

> ### Author Rebuttal · Authors · 2024-05-31
>
> Thank you for raising interesting questions! However, the reviewer’s comments suggest a mischaracterization of our work and we hope to clarify below:
>
> > Let's use a metaphor
>
> The logic of the metaphor is as follows:
> 1. LoRA acts indiscriminately on “all the model parameters”;
> 2. Hence, its side effects (if any) are also applied indiscriminately to all aspects of model behaviors, not just fairness (“blurs the whole picture”);
> 3. Hence, why do the authors (only) study fairness and why “this low-ranking should act only on fairness”?
>
> Unfortunately, this metaphor has critical flaws.
> 1. While true that LoRA is broadly applicable, it is unclear from prior work that “it has effects on every aspect of LMs”: many aspects like robustness (OOD or adversarial), privacy, and fairness are unexplored against full fine-tuning, and our work is one of the first to provide such a study.
> 2. We’re unaware of evidence for the reviewer’s claim that LoRA “has effects on every aspect of LMs” *just because* it applies to “all the parameters”.
> 3. Even if LoRA *may* have “effects on every aspect of LMs”, it does *not* mean that “this low-ranking should act only on fairness” and we made no such claim.
>
> > Provide a good reason why LoRA should have any influence on fairness
>
> (To avoid duplication, please also see our response to reviewer yDCp.)
>
> We emphasize that our main motivation is precisely that the impact on fairness is *unclear* from prior work. Our study was also in part motivated by the disparate impact of pruning and differential privacy on subgroup fairness.
>
> > Why focus on fairness? Is there any reason why this low-ranking should act only on fairness?
>
> There are indeed many aspects that we could focus on. However, fairness is an important practical aspect deserving its own investigation (as opposed to spreading our efforts to more aspects but being less comprehensive).
>
> Indeed, many other aspects may be impacted by LoRA, and that the fairness impact could be an *indirect* result these other aspects. While true, this does *not* mean that *only* fairness is impacted or worth studying, or “this low-ranking should act only on fairness”. We have made no such claims. We added LoRA and observed the actual fairness metrics end-to-end; thus while it is possible to have other mechanisms at play, our study still provides useful insight into how LoRA behaves in the real world.
>
> Nevertheless, we agree that it is interesting future work to study other potential (side) effects of LoRA.

---

> > ### Comment · Reviewer_tTvn · 2024-06-04
> > **Thank you for the explanation**
> >
> > Unfortunately, I'm not convinced of the validity of the hypothesis as, in my humble opinion, there is no causal effect between LoRa and fairness.
> > I stay with my scores.

---

> > > ### Author Response · Authors · 2024-06-05
> > >
> > > Dear Reviewer tTvn,
> > >
> > > Thank you for your engagement! We’d like to clarify that **we have not claimed a causal effect between LoRA and fairness in any part of the paper**. We hope to emphasize an important distinction between
> > >
> > > 1. Motivating the study of fairness because we hypothesized a causal effect, vs;
> > >
> > > 2. Motivating the study because fairness is not explored in the past and that there are connections to prior work that hints at a potential correlation to fairness. Again, we simply do not know if there is such a correlation (much less causation, as the reviewer is suggesting), and hence our study to find out.
> > >
> > > **It appears that the reviewer has been characterizing our work as (1), when our work is more of (2)**, as explained in Section 1 (intro) and our earlier response above.
> > >
> > > We hope that our study will be helpful for many researchers who may have different opinions on whether LoRA may have an unintended impact on fairness. Our goal is to provide experimental support so that interested researchers can use this study to either confirm or adjust their beliefs, as opposed to relying on *prior opinions* without evidence.
> > >
> > > We nevertheless acknowledge that it is the authors’ responsibility to make the above points clear and we hope to improve our presentation in the updated version.
> > >
> > > Thank you!

---

### Official Review · Reviewer_yDCp · 2024-05-11

**Rating:** 6
**Confidence:** 5
**Ethics Flag:** 1

**Summary:**

This paper studies the impact of parameter-efficient fine-tuning on the fairness of downstream tasks. The paper compares their experimental results on various settings, including text/image classification, translation, and generation tasks. They compare their results with the full-finetuning setup. The paper notes that PEFT-based fine-tuning using LORA achieves comparable performance to full fine-tuning and does not consistently excavate the unfairness of the decisions.  The paper also performs similar experiments to evaluate the calibration of the decisions and effectiveness against MIA attacks.

**Reasons To Accept:**

1. The paper is well-written and easy to follow.
2. The paper evaluates an important safety aspect of the widely used PEFT technique.
3. The paper provides an extensive set of experiments to test the hypothesis of whether PEFT-based fine-tuning negatively impacts the fairness, calibration, or privacy leakage of the resultant model.

**Reasons To Reject:**

1. The paper's premise is unclear, why should LORA worsen the fairness of the model decisions? LoRA is meant as a way to approximate full fine-tuning and with higher LORA ranks we should be able to get the exact behaviour of full-finetuning. With lower ranks, it should underfit the data -- it is not obvious to me why that should worsen the fairness? Shouldn't it just be a function of the underlying data that is being used? With this logic, shouldn't the experimental results be expected (or even obvious)? A discussion around this would be helpful.
2.  Although the paper limits itself to understanding the fairness impact of low-rank PEFT methods, it would be interesting to study the impact of other PEFT methods like adaptors, IA3, etc.

---

> ### Author Rebuttal · Authors · 2024-05-31
>
> Thank you for the thoughtful review! We respond to the main points below:
>
> > The paper's premise is unclear, why should LORA worsen the fairness of the model decisions?
>
> - We’d like to clarify that we in fact did *not* begin with a premise that LoRA would worsen fairness (indeed, it may even improve it as seen in Sec 3.2). Rather, we believe the impact on fairness is *unclear* from prior studies. This lack of clarity, together with LoRA’s wide adoption, may imply unintended and harmful consequences in high-stakes applications and has motivated our thorough investigation.
> - Our study of LoRA fairness was also motivated by the disparate impact of pruning [1] and differential privacy [2] on subgroup fairness. A common thread is that both pruning and DP reduce a model’s *fitting capacity*, which [3] argues can lead to disproportionate effects on minority data. Since LoRA also sacrifices fitting capacity for efficiency, it is natural to ask whether a similar impact exists.
>
> > Shouldn't it just be a function of the underlying data that is being used? With this logic, shouldn't the experimental results be expected (or even obvious)?
> - We respectfully disagree that fairness is always solely a simple function of data; indeed, DP training is a counter-example where the learning algorithm impacts fairness, despite the same data. We also visualized the fairness impact of subgroup sizes in **Appendix D.5**, where subgroups with fewer examples (a common factor believed to tie to fairness) do not necessarily experience worse fairness.
> - Generally, expecting results does not imply that such results were previously known. We believe a thorough study confirming (or disproving) intuitions is useful to the community.
>
> > study the impact of other PEFT methods like adaptors, IA3, etc.
> - We aim to be comprehensive on one popular method rather than to spread efforts across multiple methods. Indeed, recent work such as https://arxiv.org/pdf/2405.09673 suggests that LoRA’s wide-spread use warrants a focused study.
> - Nevertheless, we acknowledge this important future work in concluding remarks.
>
> We hope to make these discussions clearer in the updated version!
>
> [1] Tran et al., “Pruning has a disparate impact on model accuracy.” NeurIPS 2022.
>
> [2] Bagdasaryan et al., "Differential privacy has disparate impact on model accuracy." NeurIPS 2019.
>
> [3] Feldman & Zhang, "What neural networks memorize and why: Discovering the long tail via influence estimation." NeurIPS 2020.

---

### Official Review · Reviewer_Hiqr · 2024-05-11

**Rating:** 7
**Confidence:** 3
**Ethics Flag:** 1

**Summary:**

This paper shows empirical evaluations of the LoRA model on various downstream tasks by considering fairness across different sub-groups. The experimental results are informative and the conclusion that “LoRA rank has little impact on subgroup fairness” is interesting.

**Reasons To Accept:**

A detailed and comprehensive evaluation

**Reasons To Reject:**

For a draft focusing on evaluation, I think the included evaluation is sufficient.

---

> ### Author Rebuttal · Authors · 2024-05-31
>
> We thank the reviewer for their time and are glad that the reviewer finds our work valuable! Trustworthiness aspects (such as fairness) of popular algorithms are often overlooked by practitioners and we believe a thorough empirical investigation can help inform the community and future applications.

---

### Decision · Program_Chairs · 2024-07-10

**Decision:**

Accept

**Comment:**

The paper presents a study on LoRA fairness implications. The paper presents solid work with minor concerns surfaced during the review process adequately addressed during author rebuttal. While reviewer tTvn provided a different opinion, it seems they may have not understood the final aim of the paper. I recommend the authors to include reviewer recommendations in their final version.

[At least one review was discounted during the decision process due to quality]